# A Simple and Easily Implemented Method for the Regioselective Introduction of Deuterium into Azolo[1,5-*a*]pyrimidines Molecules

**DOI:** 10.3390/molecules28062869

**Published:** 2023-03-22

**Authors:** Gevorg G. Danagulyan, Henrik A. Panosyan, Vache K. Gharibyan, Ani H. Hasratyan

**Affiliations:** 1Laboratory of Bioactive Azaheterocycles, Institute of Biomedicine and Pharmacy, Russian-Armenian University, Hovsep Emin Str. 123, Yerevan 0051, Armenia; vache.gharibyan@mail.ru (V.K.G.); ani_hasratyan@mail.ru (A.H.H.); 2Scientific and Technological Center of Organic and Pharmaceutical Chemistry, The National Academy of Sciences of the Republic of Armenia, Azatutyan Ave. 26, Yerevan 0014, Armenia; henrikpanosyan@gmail.com

**Keywords:** H/D exchange, pyrazolo[1,5-*a*]pyrimidine, 1,2,4-triazolo[1,5-*a*]pyrimidine, methylpyrimidines, regioselectivity

## Abstract

A method for the technically easy-to-implement synthesis of deuterium-labeled pyrazolo[1,5-*a*]pyrimidines and 1,2,4-triazolo[1,5-*a*]pyrimidines have been developed. The regioselectivity of such transformations has been shown. ^1^H NMR and mass spectrometric methods have proved the quantitative nature of such transformations and the kinetics of deuterium exchange has been studied. Spectrally, at different temperatures (+30 °C, −10 °C and −15 °C), the kinetics of the process was studied both in CD_3_OD and in deuterated alkali.

## 1. Introduction

Azolo[1,5-*a*]pyrimidines derivatives, primarily pyrazolo[1,5-*a*]pyrimidine and 1,2,4-triazolo[1,5-*a*]pyrimidine, are known for their high biological activity. Several drugs containing the pyrazolo[1,5-*a*]pyrimidine backbone are used in medical practice. These include the sleeping pills Zaleplon [1,2], Indiplon and Lorediplon [3,4]; the sedative Ocinaplon [5]; the antifungal drug Pyrazophos [6]; the antiglycemic drug Anagliptin [7,8]; and the antitumor drug Dinaciclib [9,10] (Figure 1). Moreover, an entire series of pyrazolo[1,5-*a*]pyrimidine derivatives are registered, which have shown an antitumor effect [11,12,13,14,15,16,17,18,19,20], and effects on the central nervous system and serotonin receptors [21,22,23,24,25,26,27]. The same backbone of pyrazolo[1,5-*a*]pyrimidine is included in a number of derivatives that have shown activity as non-nucleoside inhibitors of HIV-1 reverse transcriptase (NNRTIs) [28] and respiratory syncytial virus (RSV) [29], inhibitors of RNA-polymerase of hepatitis C virus [30], as well as having antibacterial, antifungal [31] and anti-inflammatory properties [32,33,34].

The high biological activity of pyrazolo[1,5-*a*]pyrimidines stimulated interest in the development of methods for introducing various isotopes into its derivatives. In particular, compounds labeled with the [^99m^TcN]^2+^ isotope were synthesized and studied for their biodistribution in mice with tumors [35]. A 5-methylpyrazolo[1,5-*a*]pyrimidine derivative containing the ^18^F fluorine isotope was used as an imaging marker for positron emission tomography (PET) to detect a tumor [36]. It is known that deuterium-labeled compounds are also used as markers for studying the mechanisms of chemical reactions, as well as in biological research and medicinal chemistry [37,38].

Deuterated derivatives of known drugs differ from the drugs themselves by prolonging the half-life, which leads to an increase in the interval between taking the drug and, consequently, an increase in the effectiveness and safety for the patient [39].

Various methods for introducing a deuterium isotope into molecules of organic compounds are described. Formally, these methods can be divided into 3 groups, namely, proceeding under the conditions of acid or base catalysis and metal catalysis [40]. In the case of metal catalysis, Pd, Ni, Ir, Pt, Ru or their salts were used as metals. These reactions require special conditions. They were carried out under heating [41], under pressure [40,42], in a microwave oven [43], by passing pure deuterium [40], or in deuterated solvents, more often D_2_O. Deuterium atoms were introduced by H/D exchange into α-aminopyridine derivatives by heating them to 80 °C in a solution of K_2_CO_3_ in D_2_O [44]. Furthermore, by reaction with substituted acetylenes, the resulting deuterium derivatives were cyclized to pyrazolo[1,5-*a*]pyridine derivatives containing deuterium atoms. The reaction of the same pyridines with acetonitrile in basic D_2_O solution resulted in 1,2,4-triazolo[1,5-*a*]pyridine derivatives containing deuterium atoms.

Another method for introducing deuterium into an azine molecule has been described for the preparation of pyrimidine derivatives containing a deuterium atom. This involves a multicomponent cyclization of amidine, benzaldehyde and deuterated triethylamine-D_15_ in the presence of iodine at 150 °C [45]. An example of the metal catalysis used for introducing deuterium into a pyrimidine molecule is a reaction under microwave-promoted conditions catalyzed by Ruthenium(II)–Carboxylate [46].

It is also important to note that the drug Austedo (deutetrabenazine) (Figure 2), which contains deuterium atoms, is already being used in medical practice [47].

## 2. Results and Discussion

From the above, it is clear that the methods used for obtaining deuterated derivatives are not always simple. Therefore, the development of new, easily implemented methods for introducing deuterium atoms into molecules of organic substances is of interest. This is especially important in the synthesis of deuterium-containing bioactive substances. Considering the biological activity and importance of Azolo[1,5-*a*]pyrimidine derivatives in medicine, interest in developing new methods for introducing deuterium atoms into their molecules is of practical interest and is undoubtedly relevant.

This communication is devoted to a simple, easily and rapidly implemented method for introducing deuterium atoms into the molecules of substituted pyrazolo[1,5-*a*]- and 1,2,4-triazolo[1,5-*a*]pyrimidines. It is also essential that the proposed method is regioselective.

When drops of a preliminarily prepared solution of CD_3_ONa in CD_3_OD are added to the solution of 6-acetyl-2,7-dimethylpyrazolo[1,5-*a*]- (**1**) and 6-acetyl-7-methyl-2-phenylpyrazolo[1,5-*a*]pyrimidine (**2**) in CD_3_OD (Figure 1), the disappearance of the signals of two methyl groups is almost immediately observed in the ^1^H NMR spectrum. Comparison of the spectra of substances **1** and **2**, as well as the NOESY study of the spectrum of compound **1**, indicates the H/D exchange of hydrogen atoms of the methyl (7-CH_3_) and acetyl (6-COCH_3_) groups (Figure 3). Thus, in the spectrum of the deuterated product of compound **1**, the signal of the methyl groups is retained, for which the NOESY spectrum shows a response between the proton signal of one of the methyl groups (2-CH_3_) and the aromatic proton (3-H). Note that such an interaction is impossible for other methyl groups, which unambiguously proves that the 7-CH_3_ and 6-COCH_3_ protons located in the pyrimidine ring, and not the 2-CH_3_ protons located in the pyrazole ring, undergo isotopic exchange.

In order to confirm the deuterium exchange that had taken place (after the addition of CD_3_ONa), we isolated substance **3** from the NMR ampoule and compared its mass spectrum with the spectrum of compound **1** before deuterium exchange (Figure 4). As a result, the mass of product **3**, as expected, was 6 units higher than the mass of the initial substance **1** (respectively, 195 and 189). This confirms the H/D exchange in the two methyl groups.

A similar exchange of protons of two methyl groups was also observed in the ^1^H NMR spectrum of 6-acetyl-7-methyl-1,2,4-triazolo[1,5-*a*]pyrimidine (CD_3_ONa solution in CD_3_OD) (Figure 2). Almost immediately after the addition of alcoholate-D_3_ to the NMR ampoule, the signals of both methyl groups (7-CH_3_ and COCH_3_) disappeared.

Under standard conditions for recording NMR spectra, i.e., at a temperature of +30 °C, it was impossible to study the kinetics of isotope exchange because of the very high transformation rate. Therefore, we tried to study the reaction at low temperatures. According to the results of preliminary studies, the optimal temperature for this was −10 °C (Table 1, Figure 5). The rate of proton exchange in the two methyl groups was different under these conditions. The intensity of the signal of one of the methyl groups (chemical shift 3.2 ppm) (−10 °C) decreases by 6% by 5 min after the addition of deuterated sodium methoxide under these conditions, by 15 min the exchange is 22%, by 20 min the exchange reaches almost 30% and is practically completed after 2 h of measurements. According to the results of the measurements, the rate of exchange of protons of the second methyl group (chemical shift 2.28 ppm) under these conditions turned out to be significantly lower. Thus, we recorded the first results of isotope exchange only 20 min after the start of the measurements, and by 120 min only 20% of the protons in this group had undergone exchange.

In the ^1^H NMR spectrum of triazolopyrimidine **5**, recorded by the NOESY method (Figure 6), only one of the methyl groups located in the region of 2.28 ppm has a cross peak with an aromatic proton (9.3 ppm). This methyl group corresponds to the acetyl group occupying position 6. Based on this, we concluded that in the examples described, the protons of the 7-CH_3_ methyl group (3.17 ppm) are exchanged faster. We obtained similar confirmation when studying the spectrum of pyrazolopyrimidine **2** and, consequently, compound **1**.

A fast H/D exchange of protons of two methyl groups upon addition of CD_3_ONa to a solution of compound **1** in CD_3_OD was observed. Due to the rapidity of the reaction, the kinetics of the process at a temperature of −10 °C could not be fixed. However, it was registered at a lower temperature (−15 °C). Therefore, the exchange of both methyl groups approached 50% after 2 min and was practically completed within a few minutes (Table 2, Figure 7).

When studying the deuteroexchange of the 3-pyrazolyl derivative of pyrazolopyrimidine **7** containing 4 methyl groups, after adding CD_3_ONa to a solution, protons of two methyl groups in the pyrimidine ring are first exchanged (Figure 3). In the side pyrazole ring, only the signal of one of the methyl groups disappears. In this case, a cross peak was noted in the NOESY spectrum (Figure 8), indicating the interaction of the methyl group of the acetyl fragment with the aromatic proton of the pyrazole ring, which made it possible to show that the hydrogen atoms of the acetyl group of the pyrazole ring undergo exchange. Interestingly, in this example, the exchange also partially affects the aromatic protons of pyrazolo[1,5-*a*]pyrimidine.

In the study of deuterium exchange in 2-substituted 7-methyl-5-ethoxycarbonylpyrazolo[1,5-*a*]pyrimidines (**10**, **11**), after adding CD_3_ONa to the solution, a rapid exchange of hydrogen atoms in the pyrimidine ring for deuterium atoms was immediately noted. However, in these cases, the reaction was accompanied by another transformation, which was also recorded spectrally.

Thus, when comparing the ^1^H NMR spectra of 2,7-dimethyl-5-ethoxycarbonylpyrazolo[1,5-*a*]pyrimidine (**10**) recorded in CD_3_OD at temperatures of +30 °C and −10 °C, an unusual and at first glance inexplicable difference is noted. At minus temperature (−10 °C), the spectrum of the compound corresponds to the expected one and includes the signals of two methyl (2-CH_3_ 2.55; 7-CH_3_ 2.84 ppm) and one ester (OCH_2_CH_3_ 4.46, OCH_2_CH_3_ 1.44 ppm) groups, as well as singlets of two protons 3-H and 6-H (respectively, 6.70 and 7.48 ppm). In the spectrum of the same compound, recorded at +30 °C, the signals of all groups with the corresponding integrals are preserved, but two identical pairs of proton signals of two ethyl groups are observed (two quartets—OCH_2_CH_3_ 4.46 and DOCH_2_CH_3_ 3.62 ppm, each of which corresponds to one proton, and two triplets—respectively, 1.44 and 1.19 ppm, 1.5 H each) (Figure 9).

The study of the spectra recorded in CD_3_OD at −10 °C, that is, before adding CD_3_ONa to the ampoule, showed that over time there is a gradual decrease in the signals of the protons of the ethyl group and the proportional appearance of a new pair of signals of another ethyl group. In this case, the signals of all other groups remain unchanged in the spectra (Figure 7). Ultimately, the signals of the ethyl group of the original molecule, namely, those noted in the region of 4.5 ppm (CH_2_) and 1.48 ppm (CH_3_) completely disappear and are replaced by ethyl group signals that have chemical shifts in a stronger field, respectively, in the region of 1.2 (CH_3_) and 3.6 ppm (CH_2_).

We believe that the observed dynamic change in the NMR spectra is explained by the ongoing transesterification (Figure 4). In this case, the solvent molecules (CD_3_OD) interact with the ethoxycarbonyl group displacing the ethoxy group, resulting in the formation of a new 7-methyl-2-methyl-5-d_3_-metoxycarbonylpyrazolo[1,5-*a*]pyrimidine. Ethanol (C_2_H_5_OD) is formed in the solution, the signals of the groups of which are fixed in the ^1^H NMR spectrum in the form of a new ethyl group. 

At +30 °C, after adding one drop of deuterated sodium methoxide (CD_3_ONa) to the NMR ampoule, both processes rapidly occur—transesterification and H/D exchange, a result of which the signal of one of the methyl groups (7-CH_3_) disappears in the spectrum, after which 3D-methyl 2-methyl-7-(d_3_-methyl)-5-ethoxycarbonylpyrazolo[1,5-*a*]pyrimidine is immediately formed.

At low temperatures (−10 °C), we were able to study the kinetics of the entire deuterium exchange. As the experiment showed, in the beginning, transesterification already begins in CD_3_OD, which is completed even without the addition of CD_3_ONa. However, when d_3_-sodium methoxide is added, transesterification is activated, since there are no signals from the ester group of the starting ester in the spectrum. Under the same conditions (−10 °C), the kinetics of the H/D isotope exchange was studied by NMR spectral (Table 3, Figure 10).

A similar isotopic exchange, together with transesterification, was also noted for 2-phenyl-7-methyl-5-ethoxycarbonylpyrazolo[1,5-*a*]pyrimidine **11**. Thus, in the spectrum of compound **11**, which contains a phenyl group in the pyrazole ring, two processes easily occur not only in a solution of deuterated alkali (CD_3_ONa in CD_3_OD), but also in a solution of CD_3_OD: the deuterium exchange of protons of the methyl group and transesterification with the formation of d_1_-ethanol (CH_3_CH_2_OD). In the ^1^H NMR spectrum at a temperature of +30 °C after the dissolution of the substance, the signals of the protons of the ethyl group of the ester in the regions of 1.46 (t, CH_3_) and 4.47 (q, CH_2_O) almost completely disappear, and the signals of the ethyl group of the formed d_1_-ethanol CH_3_CH_2_OD (1.18—t, CH_3_ and 3.61—q, CH_2_O) become the main signals. In this case, the proton signals of all other groups of compound **11** (phenyl and methyl groups, as well as hydrogen atoms directly connected to the pyrimidine and pyrazole rings) are observed in the spectrum without changes. After adding 1–2 drops of CD_3_ONa solution in CD_3_OD to the NMR ampoule containing a solution of compound **11** in CD_3_OD, the 7-CH_3_ signal almost completely and immediately disappears (the exchange is approximately 80%), and the proton signals in the weak field (C_6_H_5_, 3-H and 6-H) remain unchanged. The exchange of protons of the methyl group for deuterium (H/D) according to the ^1^H NMR spectrum data is practically completed by 20 min after the addition of sodium d_3_-methoxide.

N-Alkylation of the pyrazolo[1,5-*a*]pyrimidine skeleton leads to a significant change in the process of deuterium exchange of methyl groups noted by us. This process was studied spectrally using the example of two salts of 6-acetyl-2,7-dimethylpyrazolo[1,5-*a*]pyrimidine—iodomethylate and iodoethylate. Namely, 6-acetyl-2,4,7-trimethylpyrazolo[1,5-*a*]pyrimidinium (**16**) and 6-acetyl-2,7-dimethyl-4-ethylpyrazolo[1,5-*a*]pyrimidinium (**17**) iodides. It is important to note that the position of the N-alkyl groups was confirmed by ^1^H NMR spectral using the NOESY technique. Thus, in the spectrum of **16** iodide, NOE (Nuclear Overhauser Effect) was noted between the protons of the N-methyl group (4.43 ppm) and the protons of the pyrazole (3-H) and pyrimidine (5-H) rings (7.25 and 9.95 ppm, respectively). The cross-peaks interactions of the 5-H proton of the pyrimidine ring with the signals of two adjacent positions of the groups, methyl N-CH_3_ and acetyl (COCH_3_), are also clearly visible. Therefore, based on the above, it was unequivocally determined that alkylation occurs at the N-4 nitrogen atom of the pyrimidine ring.

Similar interactions of protons were also noted in the NOESY study of iodoethylate **17**. It is noteworthy, that, in this case, only the protons of the N-methylene group participate in the interaction with the neighboring 3-H and 5-H protons.

In the ^1^H NMR spectrum of iodide **16** recorded in CD_3_OD without the addition of CD_3_ONa methylate, an H/D exchange of one of the methyl groups was noted. Instead of the expected signals of four methyl groups, the signals of only three of them were fixed in the spectrum (**18**) (Figure 5). Since the signal of the N-methyl group, as a rule, appears in a weaker field (in this case, it is 4.41 ppm), and the signal of the methyl group of the pyrazole ring is usually in the strongest field, the signal in the 2.78 ppm region could correspond to either a methyl group in position 7 or an acetyl group. Based on the NOESY study carried out in DMSO-d_6_ solution, it was concluded that the methyl group signal was present in the spectrum of the salt **16** in DMSO-d_6_ in the region of 3.3 ppm and disappeared due to deuterium exchange in the spectrum registered in CD_3_OD corresponding to 7-CH_3_.

Thus, methylation of the pyrimidine ring, leading to an increase in its electrophilicity, facilitates the nucleophilic isotopic exchange of hydrogen atoms of the 7-methyl group, resulting in a rapid H/D exchange even in CD_3_OD. It is interesting that, in contrast to the above examples of deuterium exchange of non-alkylated at the nitrogen atom Azolo[1,5-a]pyrimidines (**1**, **2**, **5**, **7**, **10**, **11**), with the addition of sodium d_3_-methoxide, the subsequent exchange of hydrogen atoms of the methyl fragment of the acetyl, or any other group, is not observed for only several minutes, but also during the first two days. Only by the third day does a slight decrease in the signal of the hydrogen atoms of the acetyl group (by about 25%) become noticeable in the spectrum; however, new signals begin to appear, which indicates the occurrence of other processes. The possibility of destruction at this stage should be excluded, since, in the spectrum for several more days of observations, in parallel with the decrease in the integral of the signal of the protons of the acetyl group, the signals of the remaining protons of the initial molecule (N-Me, 2-Me, 3-H and 5-H) are practically unchanged.

In case of N-ethylpyrazolo[1,5-*a*]pyrimidinium iodide **17**, as well as N-methyl derivative **16**, in the spectrum registered in CD_3_OD, i.e., without the addition of CD_3_ONa, H/D exchange occurs immediately (Figure 5) with the formation of deutero-substituted compound **19**. In the same way, i.e., the NOESY study, it was shown that 7-CH_3_ hydrogen atoms undergo rapid exchange. Further monitoring for two days did not register the H/D exchange of any other protons in the molecule.

The spectra of 6-acetyl-3,7-dimethyl-1,2,4-triazolo[1,5-*a*]pyrimidinium iodide **20** recorded in DMSO-d_6_ and CD_3_OD were identical. This indicates that the isotopic exchange does not proceed in this case in the CD_3_OD solution. Consequently, the displacement of the N-alkyl group into the triazole ring, that is, its removal from the 7-Me group, led to a decrease in the effect on potential (expected) isotopic exchange. The addition of alcoholate (CD_3_ONa) to the solution leads to the appearance of many new signals, possibly due to the opening of the pyrimidine ring and subsequent destruction of the molecule (Figure 6).

We studied a similar interactions of two more 1,2,4-triazolo[1,5-*a*]pyrimidinium iodides with solutions of deuterated sodium methoxide in deuteromethanol. In particular, the NMR spectra of 3,5,7-trimethyl-1,2,4-triazolo[1,5-*a*]pyrimidinium iodide (**21**) in CD_3_OD and CD_3_ONa/CD_3_OD were studied (Figure 7). As in the case of iodide **20** in CD_3_OD, no selective isotopic exchange of C-alkyl groups was observed without the addition of alcoholate. However, when a small amount of CD_3_ONa was added to the NMR ampule, an easy, quantitative, and, most importantly, selective basic deuteroexchange of protons of both C-methyl groups of the pyrimidinium salt **22** was noted. The signals of C-methyl groups completely disappeared at room temperature. With an increase in the duration of exposure to the deuterated reagent, the signal also disappeared from one of the aromatic protons (apparently, 2-H, located in the triazole ring—in the neighborhood of the quaternized nitrogen atom).

It is important to note that, as in the above example, in the spectrum of compound **23** (3,7-dimethyl-6-ethoxycarbonyl-1,2,4-triazolo[1,5-*a*]pyrimidine) in d_4_-methanol, no isotopic exchanges were observed. However, when d_4_-sodium methoxide is added, the proton signal of the 7-methyl group disappeared completely within a few minutes, while the signals of the remaining hydrogen atoms were preserved. Note that, as in the case of compound **21**, deuterium exchange of one of the aromatic protons, presumably located in the triazole ring, occurred over time (during 10 days of monitoring) with the formation of salt **24** (Figure 8).

It should be noted that in the latter case (salt **23**), we also observed in the ^1^H NMR spectrum a partially proceeding (significantly slower than in the case of the ester group in position 5 of compounds **10**, **11**) transesterification reaction.

Thus, on a number of examples—methyl derivatives of 1,2,4-triazolo[1,5-*a*]pyrimidinium iodides—it was confirmed that alkylation of the nitrogen atom of the triazole ring does not lead to isotopic exchange of protons of the C-methyl group located in the pyrimidine ring. Such H/D exchange requires the addition of an alcoholate (CD_3_ONa) to the medium. In the examples of pyrazolo[1,5-*a*]pyrimidinium iodides, where the nitrogen atom in the pyrimidine part of the molecule is alkylated, the H/D exchange of C-methyl groups in the pyrimidine ring is very easily realized already in a CD_3_OD solution without the addition of CD_3_ONa.

The schemes of the deuterium exchange reaction is associated with the attack of the methylate ion at the most electrophilic position in the molecule, which leads to the elimination of a proton. The resulting carbanion is stabilized by the addition of a proton (or, when the reaction is carried out in a solution of deuterated methanol, a deuterium atom).

The H/D isotopic exchange of the protons of methyl groups in all the above examples proceeds according to the mechanism of nucleophilic substitution. However, the nucleophilic substitution in bases (i.e., not salts) and alkyl iodides (i.e., azolopyrimidinium salts) occurs via different pathways and has different driving forces. In the case of the bases of **1**, **2**, **5**, **7**, **10** and **11** compounds, in a solution of CD_3_ONa in CD_3_OD, under the action of a methoxide ion, the proton of the methyl group of the pyrimidine ring is removed and replaced by a deuterium atom from the solvent molecule at the same position (Figure 9). As a result, a stepwise exchange of all hydrogen atoms of the methyl group 7-CH_3_, and then in the acetyl group, by deuterium atoms is realized.

In the case of 4-alkyl-substituted pyrazolo[1,5-*a*]pyrimidinium salts, a different driving force determines the beginning of the isotope exchange. Due to the positive charge on the nitrogen atom, the mobility of hydrogen atoms of the 7-methyl group of the pyrimidine ring increases and, as a result, already in CD_3_OD—without the addition of alcoholate—H/D exchange becomes possible (Figure 10). It should be noted that the positive charge on the nitrogen atom of the pyrimidine ring, while promoting an increase in the mobility of hydrogen atoms in the 7-methyl group, does not affect the possibility of detachment of hydrogen atoms in the acetyl group. It remains unaffected by the electronic effects of p-conjugation in the pyrimidine ring. This explains the absence of H/D exchange in the acetyl group in Azolo[1,5-*a*]pyrimidinium 4-alkyl derivatives in CD_3_OD solution.

When alcoholate is added to the reactor, azolopyrimidine is converted into its neutral form, due to the formation of NaI. The shift in the electron cloud towards the N4 atom makes the 7-position of the pyrimidine ring a nucleophilic attack target. This can lead to the opening of the pyrimidine ring and its destruction (Figure 10). Note that an ambiguous transformation of the molecule after the addition of an alcoholate was noted earlier.

The process of deuterium exchange of protons of the methyl groups of the pyrimidine ring can be explained by the relatively high CH acidity of these protons. This also correlates the shifts in the signals of these protons in a relatively weak field.

## 3. Materials and Methods

### 3.1. General Experimental Details

^1^H-, ^13^C-NMR and NOESY spectra were recorded via Varian Mercury-300 VX spectrometer (Varian, Baden, Switzerland) (^1^H-NMR 300 MHz, ^13^C-NMR 75 MHz) in a CD_3_OD at temperatures of 253, 258 and 298 K. Elemental analysis was performed via Eurovector EA 3000 instrument. Melting points were measured on instruments for determining the melting point of organic substances SMP 11 (STUART) and SMP 30 (STUART, Wickford, UK). The purity and identity of the substances were confirmed on a high-performance preparative liquid chromatograph SENMIPREPARATIV HPLC (HPLC Knauer AZURA PREP + Analitical UV Detector),Germany), as well as TLC on Silufol (UV-254). The synthesis of non-deuterated compounds **1**, **2**, **5**, **10**, **11**, **16** and **17** was carried out according to the previously described procedures [48,49]. 5-Amino-1H-pyrazole-4-carbohydrazide used for the synthesis of 3-pyrazolyl derivative of pyrazolopyrimidine **7** was purchased from Aurora Fine Chemicals LLC, San Diego, CA, USA. CD_3_OD was purchased from Sigma- Aldrich, St. Louis, MO, USA. All reagents purchased commercially were used without purification.

### 3.2. Synthetic Procedures

#### 3.2.1. Synthesis of 6-acetyl-7-methyl-3-[1-(4-acetyl-5-methyl-1H-pyrazole-1-carbonyl)]pyrazolo[1,5-*a*]pyrimidine **7**

A mixture of 5-amino-1H-pyrazole-4-carbohydrazide (130 mg, 1.2 mmol) and ethoxymethylideneacetylacetone (400 mg, 2.4 mmol) in 5 mL of absolute ethanol was refluxed for 4 h with a calcium chloride tube. After solvent evaporation, the resulting precipitate was filtered, washed with diethyl ether, recrystallized from hexane and dried to give 6-acetyl-7-methyl-3-[1-(4-acetyl-5-methyl-1H-pyrazole-1-carbonyl)]pyrazolo[1,5-*a*]pyrimidine **7** as an orange solid in 70% yield. ^1^H NMR (300 MHz, DMSO/CCl_4_ 1:3, δ, ppm): 2.48 (s, 3 H), 2.76 (s, 3 H), 2.93 (s, 3 H), 3.14 (s, 3 H), 8.12 (s, 1 H), 8.96 (s, 1 H), 9.28 (s, 1 H). ^13^C NMR (75 MHz, DMSO/CCl_4_ 1:3, δ, ppm): 12.4, 14.52, 28.82, 29.56, 103.68, 120.08, 121.8, 142.27, 146, 149.01, 149.8, 150.07, 152.88, 159.68, 191.86, 195.43. Calculated, %: C 59.07, H 4.65, N 21.53. C_16_H_15_N_5_O_3_. Found, %: C 59.03, H 4.70, N 21.50; mp: 217–218 ℃.

#### 3.2.2. General Procedure for the Preparation of Deutero-Substituted Azolo[1,5-*a*]pyrimidines **3**, **4**, **6**, **9**, **14**, **15**

A solution of several mg of compounds **1**, **2**, **5**, **7**, **10**, **11** in CD_3_OD was prepared in an NMR ampoule and ^1^H NMR spectra were recorded. Next, 2 drops of a pre-prepared solution of CD_3_ONa in CD_3_OD were added to the ampoule and the dynamics of the proton deuterium exchange in the ampoule was monitored by registering changes in the ^1^H NMR spectra (Table 4).

## 4. Conclusions

We have developed an efficient protocol for the synthesis of deuterium-labeled pyrazolo[1,5-*a*]pyrimidines and 1,2,4-triazolo[1,5-*a*]pyrimidines. It is important that the method is regioselective and leads to the introduction of deuterium atoms into the methyl group of the pyrimidine fragment of the molecule. The reaction is technically easy to implement and it can be used for labeling for biological research and studying the mechanisms of chemical reactions. It can be assumed that the method will be extended to introduce a tritium label into pharmaceuticals for use in medicine.

## Data Availability

The data are included in the article and the Appendix A.

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
