# Peer review of "A Simple and Easily Implemented Method for the Regioselective Introduction of Deuterium into Azolo[1,5-a]pyrimidines Molecules"

_molecules, 2023, doi:10.3390/molecules28062869_

Round 1

Reviewer 1 Report

Structure of pyrazolo [1,5-a]pyrimidine and 1,2,4-20 triazolo [1,5-a]pyrimidine containing drugs and Austedo should be shown.

Authors should explaine regioselectivity of the reaction (why the exchange occurred only those positions)

Authors should isolate the pure deuterated compounds 3, 4, 6, 9, 14, 15  and charaterize by 1H-NMR, 13C-NMR, 2H-NMR and HRMS.

Assign position of each proton in 1H-NMR spectra.

Author Response

Dear Reviewer,

Thank you for taking the time to review our manuscript. We appreciate your thoughtful and constructive comments, and we have carefully considered each of your suggestions.

We are pleased to inform you that we have made revisions to address your concerns and improve the quality of our work. Please find our responses to your comments below:

Comment 1: Structure of pyrazolo[1,5-a]pyrimidine and 1,2,4-triazolo[1,5-a]pyrimidine containing drugs and Austedo should be shown.

Answer 1: As you suggested, we included in the manuscript the structural formulas of drugs used in medicine, as well as the Austedo formula.

Comment 2: Authors should explain regioselectivity of the reaction (why the exchange occurred only those positions).

Answer 2: We have added a fragment to the manuscript explaining our ideas about the possible scheme (mechanism) of the reactions. Regarding regioselectivity, we would like to note that the regioselectivity in azolopyrimidines can be explained by the electronic density distribution in the 5-membered cycle and pyrimidine ring. In general, the 5-membered rings are more electron-rich than the pyrimidine ring. This distribution of electronic density leads to regioselectivity of the attack on the pyrimidine ring (where the protons of the methyl group are more reactive) in case of nucleophilic reactions. Indirect confirmation of the reactivity of these groups is also the region of signals of methyl groups in the 1H NMR spectra (in a weaker field).

Comment 3: Authors should isolate the pure deuterated compounds 3, 4, 6, 9, 14, 15 and characterize by 1H-NMR, 13C-NMR, 2H-NMR and HRMS.

Answer 3: As already noted in the article, using compound 3 as an example, the substance was isolated and characterized after the completion of the process (1H NMR and mass spectra of the compound have been registered). As you proposed, we have carried out the spectral studies of the isolated product 3. 13С NMR spectra also prove the occurrence of H/D exchange. In particular, the 13C NMR spectrum of compound 3 shows splitting (J (13C-2H) = 20.0 Hz) of the signals of carbon atoms directly bonded to deuterium atoms (COCD3; 7-CD3) and isotopic shift towards the strong field (29.9→29.2; 15.3→14.7, respectively). As well as the spectrum showed the expansion of the signals of C atoms adjacent to the CD3 groups (COCD3; C-7). In the attached documents you will find the corresponding 13C NMR spectra. We want to thank you for this important comment, which will allow us to further study in more detail the spectral transformations occurring in the process of deuterium exchange. However, in this article, we do not consider it appropriate to dwell on these studies in more detail, since this may affect the overall picture of the entire article. In addition, in some cases (eg compound 4, 2-Ph) the 13C NMR assay that we tried to do proved difficult due to the low solubility of the compound in CD3ONa/COCD3.

Comment 4: Assign position of each proton in 1H-NMR spectra.

Answer 4: The NMR spectra in the manuscript have been presented in accordance with the journal template. However, on your recommendation, we assigned the positions of protons in the spectra of all starting substances (that is, before the start of deuterium exchange). Comparison with the chemical shifts of the signals in the spectra of deuterium derivatives (including those recorded in the study of kinetics) makes it possible to note the signals of protons undergoing exchange.

All added fragments are colored yellow in the revised version of the manuscript.

Once again, we would like to express our gratitude for your valuable feedback, which has undoubtedly strengthened our work and allowed us to look at our results from a different perspective. We hope that you find our revisions satisfactory and look forward to your feedback on the revised manuscript.

Sincerely,

Gevorg Danagulyan

Reviewer 2 Report

The development of a simple way to achieve the regioselective introduction of deuterium into a potential bioactive molecule is highly important. The reaction of the method developed in this manuscript is not complicated which may be very helpful for future applications. The manuscript is well organized and I personally think it will be interesting to readers in related research field. It could be accepted in current status. If possible, I suggested the authors to provide more details for the mechanism of the regioselectivity.

Author Response

Dear Reviewer,

Thank you for taking the time to review our manuscript. We appreciate your thoughtful and constructive comments, and we have carefully considered each of your suggestions.

We are pleased to inform you that we have made revisions to address your concerns and improve the quality of our work. Please find our response to your comment below:

Comment: Provide more details for the mechanism of the regioselectivity.

Answer: We have added a fragment to the manuscript explaining our ideas about the possible scheme (mechanism) of the reactions. Regarding regioselectivity, we would like to note that the regioselectivity in azolopyrimidines can be explained by the electronic density distribution in the 5-membered cycle and pyrimidine ring. In general, the 5-membered rings are more electron-rich than the pyrimidine ring. This distribution of electronic density leads to regioselectivity of the attack on the pyrimidine ring (where the protons of the methyl group are more reactive) in case of nucleophilic reactions. Indirect confirmation of the reactivity of these groups is also the region of signals of methyl groups in the 1H NMR spectra (in a weaker field).

All added fragments are colored yellow in the revised version of the manuscript.

Once again, we would like to express our gratitude for your valuable feedback, which has undoubtedly strengthened our work. We hope that you find our revisions satisfactory and look forward to your feedback on the revised manuscript.

Sincerely,

Gevorg Danagulyan

Reviewer 3 Report

In the present manuscript entitled “A simple and easily implemented method for the regioselective introduction of deuterium into azolo [1,5-a]pyrimidines molecules”, authors have developed an efficient protocol for the synthesis of deuterium-labeled pyrazolo [1,5-a]pyrimidines and 1,2,4-triazolo [1,5-a]pyrimidines. It would be significant in medicinal chemistry and allows regioselective introduction of deuterium atoms into the methyl group of the pyrimidine fragment of the molecule.  Hence it is reasonable to accept in Molecules. I suggest authors to keep the structure of the compounds in the spectra. (Easy for the readers)

Author Response

Dear Reviewer,

Thank you for taking the time to review our manuscript. We appreciate your thoughtful and constructive comments, and we have carefully considered each of your suggestions.

We are pleased to inform you that we have made revisions to address your concerns and improve the quality of our work. Please find our response to your comment below:

Comment: Keep the structure of the compounds in the spectra.

Answer: As you suggested, we included in the manuscript the structure of the compounds in the spectra. We agree that it visually facilitates perception for the readers.

All added fragments are colored yellow in the revised version of the manuscript.

Once again, we would like to express our gratitude for your valuable feedback, which has undoubtedly strengthened our work. We hope that you find our revisions satisfactory and look forward to your feedback on the revised manuscript.

Sincerely,

Gevorg Danagulyan